# The association of gender and persistent opioid use following an acute pain event: A retrospective population based study of renal colic

Melanie Jaeger[1], Greg W. Hosier[2], Thomas McGregor[2], Darren Beiko[2], Sarah Medina Kasasni[2], Christopher M. Booth[3,4], Marlo Whitehead[5], D. Robert Siemens[2,3] *

1 Department of Anesthesiology and Perioperative Medicine, Queen's University, Kingston, Ontario, Canada, 2 Department of Urology, Queen's University, Kingston, Ontario, Canada, 3 Department of Oncology, Queen's University, Kingston, Ontario, Canada, 4 Division of Cancer Care and Epidemiology, Queen's University Cancer Research Institute, Queen's University, Kingston, Ontario, Canada, 5 ICES Queen's, Queen's University, Kingston, Ontario, Canada

* robert.siemens@kingstonhsc.ca

## Abstract

### Introduction

This study aims to explore gender-related differences in persistent opioid use following an acute pain episode and evaluate potential explanatory variables.

### Methods

This retrospective population-based study using administrative databases included all opioid-naïve patients in Ontario with renal colic between 2013 and 2017. The primary outcome was to assess any association between persistent opioid use at 3–6 months by ender. Key confounding covariates and explanatory variables examined included both care- and patient-related factors, specifically past evidence of mental health diagnoses.

### Results

The dataset of 64,240 males and 37,656 females demonstrated that 8.7% of males and 9.6% of females had evidence of persistent opioid use 3–6 months after presentation (OR 1.11, 95% CI 1.05, 1.17). Females had a higher incidence of mental health services utilization [44.5% vs 29.6% (p<0.001)] and were more likely to be on a provincial disability program [5.1% vs 3.8% (p<0.001)]. Age, income quintile, mental health diagnoses and dose of opioid prescribed were associated with the primary outcome in both genders. On adjusted analysis for multiple confounding and explanatory variables, females were still more likely than males to demonstrate persistent opioid use (OR 1.07, 95% CI 1.01, 1.13) with even more pronounced associations at 1–2 years.

**Data Availability Statement:** The risk-reduced data necessary to replicate the study are available in the manuscript. However, the authors cannot

disclose individual level data and have included group level data in the paper and tables. This is part of the agreement with ICES. The dataset from this study is held securely in coded form at ICES. While data sharing agreements prohibit ICES from making the dataset publicly available, access may be granted to those who meet pre-specified criteria for confidential access, available at https://www.ices.on.ca/DAS The full dataset creation plan and underlying analytic code are available from the authors upon request. Questions on ICES Privacy policy can be directed to privacy@ices.on.ca. Interested researchers can contact das@ices.on.ca regarding data access. The data underlying the results presented in the study are also available from the Institute for Clinical Evaluative Sciences (ICES) who can be contacted at https://www.ices.on.ca/About-ICES/ICES-Contacts-and-Sites/contact.

**Funding:** This study was supported by the Institute for Clinical Evaluative Sciences (ICES), which is funded by an annual grant from the Ontario Ministry of Health and Long-Term Care (MOHLTC). The authors gratefully acknowledge PSI Foundation for financial support for this study. No other specific funding for this project was received.

**Competing interests:** No authors have a competing interest.

**Abbreviations:** SES, socioeconomic status; NMS, Narcotic Monitoring System; OHIP, Ontario Hospital Insurance Plan; ODSP, Ontario Disability Support Program; FP, family practitioner; ED, emergency department; OME, oral morphine equivalent; PCNL, percutaneous nephrolithotomy; SWL, shockwave lithotripsy; URS, ureteroscopy; ESWL, extracorporeal shock wave lithotripsy; OR, odds ratio; CI, confidence interval.

## Interpretation

After controlling for key covariates, females are at slightly higher risk of demonstrating long term opioid use following an episode of renal colic. Evidence of prior mental health service utilization and acute colic care did not appear to significantly explain these observations.

## Introduction

Our knowledge of sex- and gender-related differences in the pathogenesis, care delivery and response to treatment is relatively nascent. Pain management represents one field that has generated significant interest in recent years [1–4]. The majority of this research demonstrates that females report more pain and are at increased risk for chronic pain. Furthermore, there is some evidence suggesting that females may experience more severe acute pain with greater pain sensitivity, enhanced pain facilitation and reduced pain inhibition [5]. Any relationship between sex, gender and pain is complex and biases in pain management are well described: influenced by both the patient and health care providers [6]. Additional research is required to elucidate mechanisms driving these differences, better understand clinical ramifications and develop strategies to address these disparities.

One critical implication of any gender-related differences in pain would appear to relate to opioid use, misuse and addiction in the general public. The opioid crisis has serious social and economic repercussions leading to multiple countries declaring public health emergencies [7, 8]. The role of the physician in the opioid crisis is acknowledged with a significant incidence of persistent opioid use following management of an acute pain event [9–13]. In addition, certain vulnerable populations have been described to be at risk for persistent opioid use including patients of lower socioeconomic status (SES) and those with mental illness [3, 14, 15]. Investigating gender-related differences in continued opioid utilization after acute pain management appears critical given that from 1990 to 2010, deaths from prescription opioid overdose in the US increased 400% for women compared to 237% for men [16].

In a previous population-based study of patients presenting with renal colic in Ontario, we demonstrated that a majority of patients were prescribed opioids at diagnosis and of those, 9% had evidence of persistent use beyond three months [16]. Urolithiasis is an extremely painful, but limited, condition and resolution of the stone does not result in any lasting tissue damage or require rehabilitation. For these reasons, it is a useful model to further study the gender-related differences following a short course of opioids. The objective of this present study is to perform a secondary analysis of this dataset to determine if opioid-naïve female patients have unique risks of persistent opioid use. We hypothesize that females are more susceptible to persistent opioid use after an acute pain event given an increased incidence of mental health related comorbidity.

## Methods

### Study design

This population-based, retrospective cohort study explores gender-related differences in persistent opioid use in previously opioid naïve patients presenting with renal colic. The present study expands on a parent study of linked administrative databases describing all opioid-naïve patients diagnosed with renal colic between July 1, 2013 and September 30, 2017 in Ontario [16]. As of 2012, all opioid prescriptions that are dispensed in Ontario are captured in the

Narcotic Monitoring System (NMS). Administrative databases utilized were all linked through Institute of Clinical Evaluative Sciences (ICES) as previously described [16]. The accuracy of these databases in terms of quality and coding has been previously discussed [17]. Codes utilized in this study have been summarized in S1 Table and the team accessed the dataset between August 2019 and March 2020.

Exclusion criteria were patients that had been dispensed an opioid in the year prior to their renal colic presentation, those who had a renal colic episode in the year prior, and any patient who had received palliative care services either the year before or the year after their index presentation. This study was approved by the Queen's University Health Sciences and Affiliated Hospitals Research Ethics Board. All data were fully anonymized before accessing them and requirement for informed consent was waived.

## Outcomes

Persistent opioid use after renal colic by documented gender was the primary outcome. Persistent opioid use was defined by one or more opioid prescription(s) dispensed between 0 and 90 days following the index renal colic visit that subsequently resulted in one or more prescription(s) for opioids that was dispensed between 91 and 180 days following the index visit. These definitions are similar to that of other comparable studies [10, 11] although we also determined dispensed opioids 6–12 months and 1–2 years after index diagnosis. Secondary outcomes included any subsequent diagnosis of opioid overdose, opioid addiction, hyperalgesia, and opioid-related death [17, 18].

## Covariates

Covariates were determined *a priori* and conceptualized as confounding variables such as age, income quintiles, geographic location, patient comorbidities (Charlson index) [19], and enrollment with a primary care practitioner (PCP). The majority of these explanatory factors were conceived, documented on a Dataset Creation Plan, for the parent study [16] including clinical care-related covariates such as number of emergency department (ED) visits, gender of prescriber, number of PCP visits, and total oral morphine equivalents (OMEs) [20] dispensed during the index presentation. Given past literature focusing on sex and gender on pain perception, experience and management [3, 14, 15] we also included new covariates: mental health utilization, substance abuse and disability program registration (ODSP). The most important covariates not available in this dataset are the kidney stone details (size and location).

## Statistics

Descriptive statistics (ratios) were used for demographic and baseline characteristics and to compare proportions between groups the Chi-square test was used. To identify the factors associated with the primary outcome, both univariate and multivariable logistic regression models were used. All patient-related and care-related variables were included in the original models and then subsequent iterative models were performed with groupings of covariates as described. Given the previous literature around gender, mental health diagnoses and pain outcomes we performed a test of interaction terms of gender and any mental health services utilization with the primary outcome of persistent opioid use. Missing data was included in the analyses as unknown indicator variable. A two-sided p-value of <0.05 was considered statistically significant. As per institutional policy, cells with <6 patients were not reported due to privacy concerns. Data were analysed using SAS Stat 14.3.

**Table 1. Patient characteristics of male and female patients with renal colic in Ontario during 2013–2017.**

| Characteristic | Male | Female | P Value[a] |
|---|---|---|---|
| | n = 64,240 | n = 37,656 | |
| Age at index, year, n (%)[b] | | | < .001 |
| 0–18 | 754 (1.2) | 938 (2.5) | |
| 19–39 | 13,969 (21.7) | 10,867 (28.9) | |
| 40–59 | 29,988 (46.7) | 15,640 (41.5) | |
| 60–79 | 17,403 (27.1) | 8,827 (23.4) | |
| 80 + | 2,126 (3.3) | 1,384 (3.7) | |
| Neighborhood income, n (%)[b] | | | < .001 |
| 1 –Lowest quintile | 11,474 (17.9) | 7,471 (19.8) | |
| 2 | 12,572 (19.6) | 7,836 (20.8) | |
| 3 | 13,032 (20.3) | 7,530 (20.0) | |
| 4 | 13,531 (21.1) | 7,587 (20.1) | |
| 5- Highest quintile | 13,492 (21.0) | 7,161 (19.0) | |
| ODSP record, n (%)[b] | | | < .001 |
| No | 61,793 (96.2) | 35,739 (94.9) | |
| Yes | 2,447 (3.8) | 1,917 (5.1) | |
| Geographic, n (%)[b] | | | < .001 |
| Urban | 57,072 (88.8) | 32,948 (87.5) | |
| Rural | 7,070 (11.0) | 4,668 (12.4) | |
| Any mental health utilization, n (%) | 18,987 (29.6) | 16,770 (44.5) | < .001 |
| Anxiety by primary care, n (%) | 12,736 (19.8) | 11,808 (31.4) | < .001 |
| Anxiety by psychiatrist, n (%) | 3,184 (5.0) | 2,928 (7.8) | < .001 |
| Mood disorder by primary care, n (%) | 2,523 (3.9) | 2,806 (7.5) | < .001 |
| Mood disorder by psychiatrist, n (%) | 1,621 (2.5) | 1,478 (3.9) | < .001 |
| Substance abuse, n (%) | 2,429 (3.8) | 1,239 (3.3) | < .001 |
| Self-harm, n (%) | 182 (0.3) | 213 (0.6) | < .001 |
| Charlson Index, n (%)[b] | | | 0.03 |
| 0 | 60,463 (94.1) | 35,486 (94.2) | |
| 1–2 | 2,797 (4.4) | 1,672 (4.4) | |
| 3 + | 980 (1.5) | 498 (1.3) | |
| Enrolled with a Family Practice, n (%) | 52,336 (81.5) | 31,912 (84.7) | < .001 |

ODSP = Ontario Disability Support Program.

[a] Chi-square test.

[b] Column percentages, may not add to 100% due to missing or unrepresented data.

## Results

The dataset was comprised of 64,240 males and 37,656 females who presented with renal colic. The characteristics of males and females presenting with renal colic in Ontario are described in Table 1 with some differences in baseline demographic data. Males tended to be slightly older compared to females with further minor differences in SES, comorbidity and enrollment in a PCP practice. Females however, had a significantly higher incidence of mental health services utilization [44.5% vs 29.6% (p<0.001)]. There was also a higher proportion of female patients on the Ontario Disability Support Program (ODSP) in the 3 years prior to presentation [5.1% vs 3.8% (p<0.001)]. Although females appeared to have slightly earlier resolution of their renal colic event and more likely to have earlier stone related surgery, there was more documentation of primary care visits compared to males during the initial presentation (Table 2).

**Table 2. Clinical care characteristics of male and female patients with renal colic in Ontario during 2013–2017.**

| Characteristic | Male | Female | P Value[a] |
|---|---|---|---|
| | **n = 64,240** | **n = 37,656** | |
| Months in renal colic, n (%)[b] | | | < .001 |
| < 2 | 48,042 (74.8) | 28,902 (76.8) | |
| ≥2 | 16,198 (25.2) | 8,754 (23.2) | |
| ED visits during renal colic, n (%)[b] | | | 0.09 |
| 0 | 45,882 (71.4) | 26,633 (70.7) | |
| 1 | 12,976 (20.2) | 7,843 (20.8) | |
| 2 | 3,596 (5.6) | 2,136 (5.7) | |
| > 2 | 1,786 (2.8) | 1,044 (2.8) | |
| PCP visits during renal colic, n (%)[b] | | | < .001 |
| 0 | 33,733 (52.5) | 18,027 (47.9) | |
| 1 | 17,896 (27.9) | 11,219 (29.8) | |
| 2 | 7,150 (11.1) | 4,648 (12.3) | |
| > 2 | 5,461 (8.5) | 3,762 (10.0) | |
| Surgery within 6 months, n (%) | 14,920 (23.2) | 9,803 (26.0) | < .001 |
| Time to surgery n (%)[b] | | | < .001 |
| No surgery | 49,320 (76.8) | 27,853 (74.0) | |
| < 1 month | 10,021 (15.6) | 6,900 (18.3) | |
| ≥1 month | 4,899 (7.6) | 2,903 (7.7) | |
| Any opioid dispensed 0–90 days, n (%) | 43,625 (67.9) | 23,373 (62.1) | < .001 |
| Sum of opioid days supplied, n (%)[b] | | | < .001 |
| 1–2 | 10,346 (16.1) | 5,378 (14.3) | |
| 3–4 | 14,545 (22.6) | 7,763 (20.6) | |
| 5–7 | 10,220 (15.9) | 5,671 (15.1) | |
| > 7 | 8,289 (12.9) | 4,308 (11.4) | |
| Total oral morphine equivalents, n (%)[b] | | | < .001 |
| 1 - <100 | 7,334 (11.4) | 4,495 (11.9) | |
| 100 - <150 | 8,155 (12.7) | 4,895 (13.0) | |
| 150 - <200 | 9,406 (14.6) | 4,727 (12.6) | |
| 200 - <300 | 9,058 (14.1) | 4,523 (12.0) | |
| 300 + | 9,410 (14.6) | 4,453 (11.8) | |

ED = Emergency department.

PCP = Primary care practitioner.

[a] Chi-square test.

[b] Column percentages, may not add to 100% due to missing or unrepresented data.

Of those previously opioid naïve patients prescribed an opioid during their colic event, 3,794 (8.7%) males and 2,236 (9.6%) females continued to fill at least one opioid prescription 3–6 months after the initial presentation. Consequential outcomes secondary to opioid use such as addiction (0.4%), overdose (0.3% males, 0.4% females) and hyperalgesia (0.4% males, 0.6% females) were similar for both genders. However, longer follow-up of those opioid naïve patients filling a prescription at 3–6 months demonstrated higher rates of further continued use 6–12 months after the initial stone presentation in females (31.8% vs. 26% males). At 1–2 years, 38.1% of females compared with 31.4% of males continued to fill at least one opioid prescription.

The dataset demonstrated that 8.7% of males and 9.6% of females had evidence of persistent opioid use 3–6 months after presentation (OR 1.11, 95% CI 1.05, 1.17). The factors associated

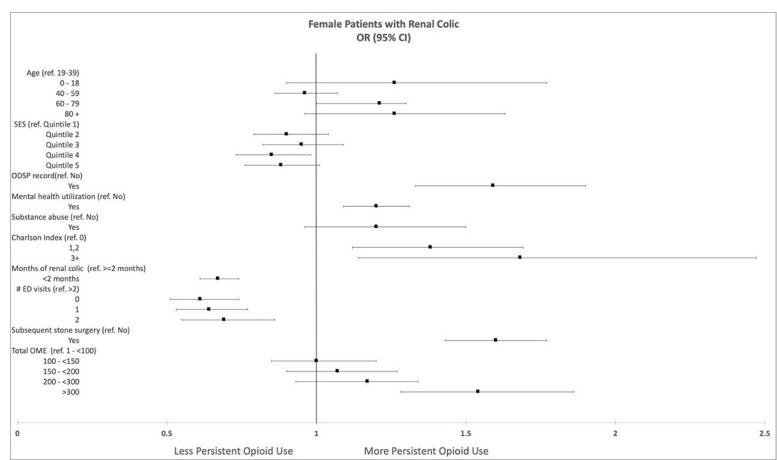
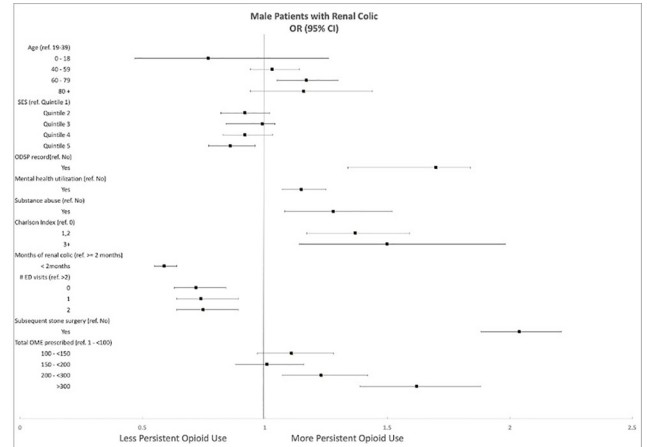

**Fig 1.** A and B. Odds ratios for the associations between characteristics and persistent opioid use by gender. Controlled in logistic regression for all covariates listed as well as enrollment in primary care practice, geographic region (urban, rural), specialty and gender of initial opioid prescriber as well as sum of opioid days supplied. OME = Oral morphine equivalents, ED = Emergency Department, ODSP = Ontario Disability Support Program, SES = Socio-economic status.

with persistent opioid use on unadjusted analyses were similar for both males and females (S2 Table). Age was associated with continued opioid use in both genders. Evidence of any mental health services utilization appeared to be associated with the primary outcome in both males (OR 1.30 95% CI 1.21, 1.39) and females (OR 1.32 95% CI 1.21, 1.44). Evidence of substance abuse and self-harm were similarly associated with persistent opioid use in unadjusted analyses. As a direct comparison, Fig 1 describes the odds ratios from multivariable models incorporating all covariates for each gender separately. In these adjusted models, mental health utilization, substance abuse and enrollment in a disability program were associated with long term opioid use to a similar degree in both genders although substance abuse was not significant in females. A test of interaction terms of gender and any mental health services utilization with the primary outcome was not significant.

On multivariable analysis of all covariates including gender, females were associated with persistent opioid use 3–6 months after initial presentation (OR 1.07, 95% CI 1.01, 1.13, p = 0.03) despite controlling for all key covariates investigated in this cohort (S2 Table). Both the duration and total amount of OME prescribed was strongly associated with persistent use. Female gender continued to be associated with the primary outcome after adjusting for each of these individual covariates. In subsequent iterative sensitivity models, incorporating confounding variables of age, SES and comorbidity and then more explanatory variables such as mental health services utilization, renal colic care and total OME originally prescribed, gender continued to be associated with persistent use (Fig 2). These results were similar if not more pronounced when extending these observations out to those filling an opioid prescription 1–2 years after the index renal colic episode (Fig 3). Unadjusted risk between genders for the primary outcome was OR 1.35 (95% CI 1.21, 1.50) and in the fully adjusted model: 1.28 (95% CI 1.14, 1.43).

## Interpretation

In this population-based study, we demonstrate gender-related differences in persistent opioid use beyond 3 months after an intended short-term course required for renal colic, independent of multiple key confounders or explanatory variables. Of the 2,236 (9.6%) previously opioid naïve females that continued to fill opioid prescriptions after an acute pain event, more

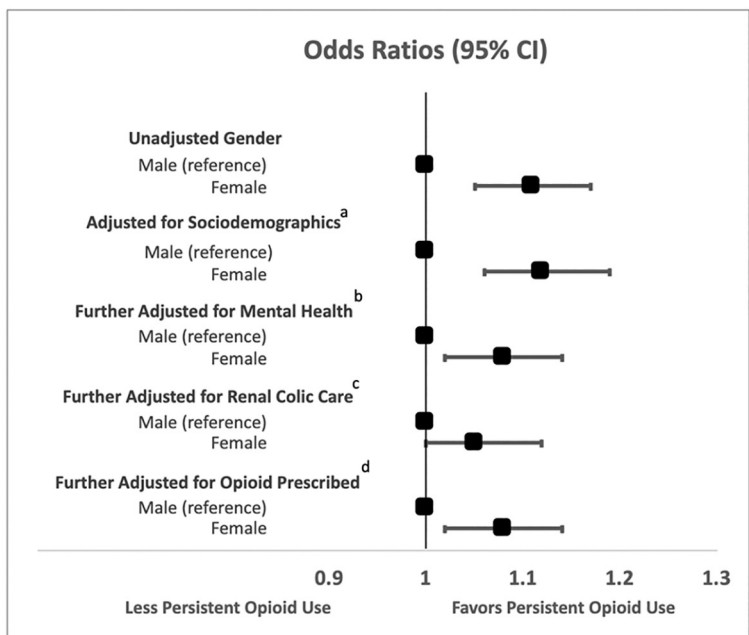

**Fig 2. Odds ratios for the associations between gender and persistent opioid use at 3–6 months after index.** [a] Adjusted in logistic regression for age, socio-economic status, comorbidity [b]Further adjusted for mental health utilization, substance abuse and Ontario Disability Support Program [c]Further adjusted for time of renal colic episode, subsequent need for stone surgery, number of Emergency Department visits [d]Further adjusted for total oral morphine equivalents prescribed during renal colic management.

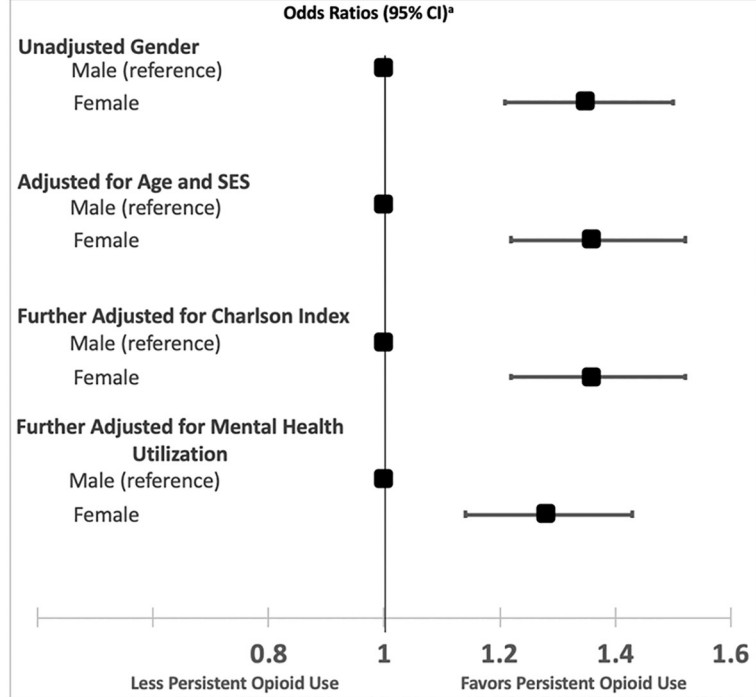

**Fig 3. Odds ratios for the associations between characteristics and persistent opioid use by gender at 1–2 years after index.** [a] Adjusted in logistic regression.

than a third had evidence of persistent opioid use 1–2 years later. Females were more likely to have documented utilization of mental health services compared to males in this cohort and given the strength of its association with persistent opioid use, it could be hypothesized as an explanatory factor although gender continued to be significant even after adjusting for this covariate. Key treatment-related factors associated with persistent opioid use including need for surgical intervention and the amount of opioids dispensed appeared to be similarly associated in both males and females.

Female patients in this cohort, representing all those diagnosed with renal colic during the study period, differed in several parameters compared to their male counterparts, some of which have been previously associated with continued opioid utilization. Female patients tended to be somewhat younger with some evidence of lower SES, although other statistically significant variances in baseline characteristics such as rurality, comorbidity and previous stone surgery were likely not clinically different. Nonetheless, previous studies exploring associations of persistent opioid use and patient-related factors would suggest many of these variables such as age, SES and comorbidity may confound these findings [10, 11]. Female patients were less likely to be prescribed an opioid during their renal colic presentation (62.1% vs. 67.9% for men) and the total OME prescribed to women was significantly less (23.8% females prescribed >200 OME vs. 28.7% for males) which is consistent with previous studies. This finding is interesting given the strength of the association of dose and duration of original opioid prescriptions and persistent use in otherwise unselected patients with acute or chronic non-cancer pain [21]. As well, females appeared to have slightly earlier resolution of their renal colic event and more likely to have earlier stone related surgery. Despite these findings that might predict less ongoing opioid utilization, females still demonstrated more opioid use 3–6 months after the acute pain event in both unadjusted and adjusted analysis.

There were other apparent gender-related differences in the management of patients with renal colic which may have contributed to the primary outcome including more surgical interventions (23.2% for males vs. 26% for females) and more health provider visits during the initial renal colic event which may have been associated with repeated access to opioid prescriptions. We have previously reported several kidney stone surgery specific risk factors for persistent opioid use [16] including an increased number of interventions as well as surgical type. These factors appeared to be similarly associated with the primary outcome in both males and females in this present study (S3 Table).

Multiple studies have investigated differences in opioid prescribing and opioid use between genders with consistent findings that characteristics such as mental health diagnoses and post-traumatic stress are strongly associated with opioid utilization in the chronic pain setting [2, 3, 14]. Similarly, a limited number of recent studies exploring persistent opioid use following acute pain events in previously opioid-naïve patients have found similar results in terms of gender-related risk factors [21, 22]. Involvement in a disability program, such as the ODSP in this present study, has been also been shown to be associated with long term opioid use [23], but few have included such programs in their analyses of persistent opioid use after short-term acute opioid prescriptions. In this current study, mental health services utilization appeared to be strongly associated with continued opioid use in both males and females in adjusted analysis. However, the incidence of such diagnoses was significantly higher in females than their male counterparts.

Given the millions of low-complexity procedures that are performed every day, in addition to the various acutely painful conditions that present to primary care and urgent care clinics, these significant rates of persistent opioid use are concerning. The strong association of previous mental health services utilization and continued opioid use is consistent for both males and females, however the increased incidence of a mental health diagnosis in females in this

large population-based cohort emphasizes their vulnerability. The specific etiological basis underlying any gender differences in pain experiences are complex and multi-factorial, and emerging evidence suggests endogenous opioid functioning may play a causal role in these disparities, including sex hormones as factors influencing pain sensitivity [1]. Psychosocial processes, stereotypical gender roles manifesting in differences in pain expression and sex differences in responses to analgesic medications have been implicated [1, 5]. Despite adjusting for multiple key factors associated with persistent opioid use in this present study, the fact that female gender remained an independent variable associated with persistent opioid use underscores the need to consider gender when identifying vulnerable populations with regard to opioid prescribing.

This study is limited by the retrospective nature as well as the accuracy of the data used, including diagnostic codes, procedure codes and opioid prescription capture, leading to potential misclassification bias. Furthermore, it is not possible to determine potential additional diagnoses for which ongoing opioid management would be considered, or the severity of the diagnoses that were coded. In addition, the NMS only tracks opioids dispensed, not necessarily those actually utilized by patients, or opioids utilized by a patient from a non-physician source (e.g. friend/relative's prescription, street). Importantly, information around the size and location of urolithiasis was not captured in the current study. Finally, there remains the potential threat of residual confounding. Strengths of the study however include the large population-based cohort size and the ability to determine opioid utilization prior to the urolithiasis diagnosis as the NMS captures opioid prescriptions for patients of all ages in this mandatory, province-wide reporting system.

In conclusion, the majority of patients diagnosed with renal colic in this real-world study were prescribed opioids, despite increasing awareness of risks and documented efficacy of non-opioid medications. Females were at slightly higher risk to fill prescriptions beyond 3 months after the acute pain event and this remained significant even after adjustment for key confounders. The data support and highlight the urgent need for opioid reduction strategies, including opioid-reduced prescriptions and multimodal non-opioid analgesia strategies with a focus on provider education to identify vulnerable populations.

## Supporting information

**S1 Checklist.**
(DOCX)

**S1 Table. Codes utilized.**
(DOCX)

**S2 Table. Odds ratios for the association between characteristic and persistent opioid use.**
(DOCX)

**S3 Table. Odds ratios for the association between surgical factors and persistent opioid use by gender.**
(DOCX)

## Author Contributions

**Conceptualization:** Melanie Jaeger, Greg W. Hosier, Thomas McGregor, Darren Beiko, Sarah Medina Kasasni, Christopher M. Booth, Marlo Whitehead, D. Robert Siemens.

**Data curation:** Greg W. Hosier, Christopher M. Booth.

**Formal analysis:** Greg W. Hosier, Marlo Whitehead, D. Robert Siemens.

**Investigation:** Greg W. Hosier, Thomas McGregor, Darren Beiko, Sarah Medina Kasasni, D. Robert Siemens.

**Project administration:** Melanie Jaeger.

**Supervision:** Melanie Jaeger, Thomas McGregor, Christopher M. Booth, D. Robert Siemens.

**Validation:** Christopher M. Booth.

**Writing – original draft:** Melanie Jaeger, D. Robert Siemens.

**Writing – review & editing:** Melanie Jaeger, Greg W. Hosier, Thomas McGregor, Darren Beiko, Sarah Medina Kasasni, Christopher M. Booth, Marlo Whitehead.

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
