## [Decision Letter · Decision Letter 0]

25 May 2021

PONE-D-21-08670

The Association of Gender and Persistent Opioid Use Following an Acute Pain Event: A Retrospective Population Based Study of Renal Colic

PLOS ONE

Dear Dr. Siemens,

Thank you for submitting your manuscript to PLOS ONE. After careful consideration, we feel that it has merit but does not fully meet PLOS ONE’s publication criteria as it currently stands. Therefore, we invite you to submit a revised version of the manuscript that addresses the points raised during the review process.

We look forward to receiving your revised manuscript.

Kind regards,

Jingjing Qian

Academic Editor

PLOS ONE

Additional Editor Comments:

Overall comment: the manuscript entitled “The Association of Gender and Persistent Opioid Use Following an Acute Pain Event: A Retrospective Population Based Study of Renal Colic” assessed gender-related differences in persistent opioid use following an acute pain episode and evaluated potential explanatory variables. Using administrative databases including a large cohort of opioid-naïve patients in Ontario with renal colic between 2013 and 2017, the authors found that females were at higher risk of demonstrating long term opioid use following an episode of renal colic. I have the following comments for editor’s and authors’ consideration:

1. Methods: page 7, please justify the selected covariates to briefly explain the rationale of the selections.

2. Methods: in the statistics section, please describe the individual interaction terms that were tested in the multivariable logistic regression model and the rationale.

3. Results: page 10, the authors mentioned “Age was associated with continued opioid use in both genders, however females aged 0-18 had a higher odds ratio (OR) (1.14, 95% confidence interval (CI) 0.82, 1.58) whereas males aged 0-18 had a lower OR (0.77, 95% CI 0.48, 1.25).” when describing results from the unadjusted analyses. However, given that these 95% CIs indicated statistical non-significance, please modify the description to avoid presenting “higher” or “lower” odds ratios for females and males. Similarly, please also revise the corresponding content in the Interpretation section on page 13 (first paragraph).

4. Results: in Tables 1 and 2, all patient level and clinical level covariates were statistically different between females and males. An alternative (potentially better) approach is to use propensity score matching or weighting to balance these covariates before assess the association between gender and persistent opioid use following an acute pain episode. Please discuss why propensity score was not used for your analysis and if it might show different findings from your results.

5. Results: there are a few places where the authors indicated "data not shown" instead of adding corresponding results in the supplemental materials. Please make your data/results fully available to the audience.

6. Interpretation: based on the authors’ main finding of “females were associated with persistent opioid use 3-6 months after initial presentation (OR 1.07, 95% CI 1.01, 1.13, p=0.03)”, please discuss clinical significance (instead of statistical significance) of the marginal difference (less than 4% in difference based on the adjusted OR) in persistent opioid use 3-6 months between females and males.

Journal Requirements:

2.In your ethics statement in the Methods section and in the online submission form, please provide additional information about the data used in your retrospective study. Specifically, please ensure that you have discussed whether all data were fully anonymized before you accessed them and/or whether the IRB or ethics committee waived the requirement for informed consent. If patients provided informed written consent to have data from their medical records used in research, please include this information.

3. Thank you for providing the date(s) when patient medical information was initially recorded. Please also include the date(s) on which your research team accessed the databases/records to obtain the retrospective data used in your study.

"This study was supported by the Institute for Clinical Evaluative Sciences (ICES), which

is funded by an annual grant from the Ontario Ministry of Health and Long-Term Care

(MOHLTC)"

"All acknowledgments are added but this was unfunded research. "

Reviewers' comments:

Reviewer's Responses to Questions

**Comments to the Author**

1. Is the manuscript technically sound, and do the data support the conclusions?

Reviewer #1: Yes

2. Has the statistical analysis been performed appropriately and rigorously? 

Reviewer #1: Yes

3. Have the authors made all data underlying the findings in their manuscript fully available?

Reviewer #1: Yes

4. Is the manuscript presented in an intelligible fashion and written in standard English?

Reviewer #1: Yes

5. Review Comments to the Author

Reviewer #1: This manuscript is technically sound and overall easy to read. The authors should be congratulated. I have no concerns regarding the conduct of the study. Some specific recommendations regarding the write-up are included below.

Title:

-For clarity, consider revising to, "The association between gender and persistent opioid use..."

Ethics statement:

-There is no explicit statement that the Queen's IRB approved the research or what type of consent was obtained.

Data availability section:

-Please supply details regarding where/how all raw study data can be obtained (even though it seems it is publicly available).

Abstract:

-In the first sentence of the results, please indicate some statistical measure (with error term) regarding whether the unadjusted proportion of males and females with persistent opioid use are different. (Please make the same clarification in the main Results section.)

-In the conclusion, consider adding a term such as, "females are at a 'slightly' higher risk..." since the prevalences of 8.7% and 9.6% (and the OR of 1.07) are, arguably, fairly similar from a clinical standpoint.

Results:

-When discussing the prevalence of longer term opioid use, please provide statistical metrics for the comparisons of proportions.

Figure 1:

-The text in the figures is quite small. Consider splitting the figure across two pages for easier readability. Or, if color-coded, you could potentially put the male and female data on the same plot.

Figure 2:

-This is a great figure. Very digestable and informative.

Figure 3:

-Also a great figure. Consider adding an adjustment for prescription of opioids 3-6 months after index.

Other:

-Both gender and sex are mentioned in the Intro, but then the manuscript seems to focus on only gender. What was the rationale behind focusing on gender as opposed to sex (since they are not interchangeable variables)? Does the administrative database truly capture gender, sex, or both?

6. PLOS authors have the option to publish the peer review history of their article (what does this mean?). If published, this will include your full peer review and any attached files.

Reviewer #1: No

---

## [Author Response · Author response to Decision Letter 0]

8 Jun 2021

Editors and Reviewers Comments: 

Overall comment: the manuscript entitled “The Association of Gender and Persistent Opioid Use Following an Acute Pain Event: A Retrospective Population Based Study of Renal Colic” assessed gender-related differences in persistent opioid use following an acute pain episode and evaluated potential explanatory variables. Using administrative databases including a large cohort of opioid-naïve patients in Ontario with renal colic between 2013 and 2017, the authors found that females were at higher risk of demonstrating long term opioid use following an episode of renal colic. I have the following comments for editor’s and authors’ consideration:

1. Methods: page 7, please justify the selected covariates to briefly explain the rationale of the selections.

Thank you for this comment around the covariates chosen. The selected covariates and explanatory factors described herein were conceived a priori by the research team, and are documented on a Dataset Creation Plan, for the parent study which is listed as reference 16. Given past literature focusing on sex and gender on pain perception, experience and management we also included the new covariates such as mental health utilization, substance abuse and disability program registration (ODSP). The most important covariates not available in this dataset are the kidney stone details (size and location) and a limitation of the observations. 

2. Methods: in the statistics section, please describe the individual interaction terms that were tested in the multivariable logistic regression model and the rationale.

We have added this wording to the methodology: Given the previous literature around gender, mental health diagnoses and pain outcomes we performed a test of interaction terms of gender and any mental health services utilization with the primary outcome of persistent opioid use. 

3. Results: page 10, the authors mentioned “Age was associated with continued opioid use in both genders, however females aged 0-18 had a higher odds ratio (OR) (1.14, 95% confidence interval (CI) 0.82, 1.58) whereas males aged 0-18 had a lower OR (0.77, 95% CI 0.48, 1.25).” when describing results from the unadjusted analyses. However, given that these 95% CIs indicated statistical non-significance, please modify the description to avoid presenting “higher” or “lower” odds ratios for females and males. Similarly, please also revise the corresponding content in the Interpretation section on page 13 (first paragraph).

This is an excellent point and we simply removed from the manuscript as the point around direction was a minor observation from Supplementary Table 2 and not additive given the CIs with respect to the younger age groups. 

4. Results: in Tables 1 and 2, all patient level and clinical level covariates were statistically different between females and males. An alternative (potentially better) approach is to use propensity score matching or weighting to balance these covariates before assess the association between gender and persistent opioid use following an acute pain episode. Please discuss why propensity score was not used for your analysis and if it might show different findings from your results.

Thank you for this question about the adjusted models. We appreciate the academic discussions around the differences and potential benefits (and drawbacks) of PSM compared to regression analyses employed herein. In this present study, we wanted to identify potential explanatory factors for any observed differences between the genders and any association with the primary outcome and therefore utilized univariate and multivariable logistic regression models as informative for the clinical reader. It is a formidable tool to appraise multivariate predictors of dichotomous events and evaluate the independent predictive role of more than one independent variable of interest. However, we recognize that these models may not perfectly guide the strength of the confounders. 

As mentioned above, there is some imbalance between the covariates between the genders (Table 1) although we felt these were in general not clinically remarkable and only significant because of the large number of patients presenting with renal colic in the dataset (n=101,978). The benefit of PSM to control for overt, measurable biases did not resonate for us as the exposure in this study was gender (male/female) and we simply didn’t conceptualize any propensity score as the conditional probability of receiving the exposure given a vector of measured covariates. However, as conceived as a balancing variable to achieve “balance” between the two groups (male/female), especially given the potential complexity of the methodology (different propensity score matching techniques), PSM are likely superior to standard MVA methods only in small datasets (which of course is not the case in this study). It is our understanding the literature would suggest that PSM performance in larger datasets is similar or less precise than logistic regression. 

5. Results: there are a few places where the authors indicated "data not shown" instead of adding corresponding results in the supplemental materials. Please make your data/results fully available to the audience.

Thank you. We have removed that commentary in the Results as it was actually a mistake as it is included in the Supplementary Table 2. Further, we have added Supplementary Table 3 to reference the statement in the Interpretation. 

6. Interpretation: based on the authors’ main finding of “females were associated with persistent opioid use 3-6 months after initial presentation (OR 1.07, 95% CI 1.01, 1.13, p=0.03)”, please discuss clinical significance (instead of statistical significance) of the marginal difference (less than 4% in difference based on the adjusted OR) in persistent opioid use 3-6 months between females and males.

Thank you for this comment as it was important as we put this manuscript together: not just to confirm the gender differences in persistent opioid use but to dissect the “explanatory” variables associated with the outcome. Hence the reason for the different iterative (sensitivity) regression analyses rather than PSM as described above. 

Within the Interpretation we do comment on the significance of these observed differences between the genders and this addition to the literature. Although the main outcome was persistent opioid use in previously opioid naïve patients at 3-6 months we also observed more stark differences later on within the NMS data (1-2 years): 38.1% of females compared with 31.4% of males. Further, although only a small difference based on the adjusted OR, this was adjusted for really key confounders (baseline, mental health utilization, opioids prescribed, surgical variables and interactions with health professionals) and yet we still see a difference between the genders. Our interpretation was that despite controlling for these variables, females still demonstrated more opioid use 3-6 months after an acute pain event in adjusted analysis (OR 1.07 at 3-6 months and 1.28 at 1-2 years later) suggesting there may be other unmeasured confounding variables to explain the ORs or “true” differences in other bio-psycho-social determinants of pain and opioid prescription filling. In fact, we discuss in the Interpretation that females appeared to have slightly earlier resolution of their renal colic event and more likely to have earlier stone related surgery. “Despite these findings that might predict less ongoing opioid utilization, females still demonstrated more opioid use 3-6 months after the acute pain event in both unadjusted and adjusted analysis”. 

Our interpretation around the clinical relevance was that “Despite adjusting for multiple key factors associated with persistent opioid use in this present study, the fact that female gender remained an independent variable associated with persistent opioid use underscores the need to consider gender when identifying vulnerable populations with regard to opioid prescribing.”

Journal Requirements:

 We have gone over and assured formatting issues resolved. 

2.In your ethics statement in the Methods section and in the online submission form, please provide additional information about the data used in your retrospective study. Specifically, please ensure that you have discussed whether all data were fully anonymized before you accessed them and/or whether the IRB or ethics committee waived the requirement for informed consent. If patients provided informed written consent to have data from their medical records used in research, please include this information.

Done

3. Thank you for providing the date(s) when patient medical information was initially recorded. Please also include the date(s) on which your research team accessed the databases/records to obtain the retrospective data used in your study.

 Done

"This study was supported by the Institute for Clinical Evaluative Sciences (ICES), which

is funded by an annual grant from the Ontario Ministry of Health and Long-Term Care

(MOHLTC)"

"All acknowledgments are added but this was unfunded research. "

 Thank you. The statement we provide is from ICES which is supported by the Ministry of Health, but this specific project was not supported otherwise financially. We have removed this from the Acknowledgement but will add to the funding statement.

This would then be the funding statement…

“This study was supported by the Institute for Clinical Evaluative Sciences (ICES), which is funded by an annual grant from the Ontario Ministry of Health and Long-Term Care (MOHLTC). No other specific funding for this project was received.”

Thank you and we have added one supplementary table as we do feel it is likely useful for the readership. As above, no statements (data not shown) are now present within the manuscript.

Added these captions to the end of the manuscript.

Reviewers' comments:

Reviewer's Responses to Questions

Comments to the Author

1. Is the manuscript technically sound, and do the data support the conclusions?

Reviewer #1: Yes

2. Has the statistical analysis been performed appropriately and rigorously? 

Reviewer #1: Yes

3. Have the authors made all data underlying the findings in their manuscript fully available?

The dataset from this study is held securely in coded form at ICES. While legal data sharing agreements between ICES and data providers (e.g., healthcare organizations and government) prohibit ICES from making the dataset publicly available, access may be granted to those who meet pre-specified criteria for confidential access, available at www.ices.on.ca/DAS (email: das@ices.on.ca). The full dataset creation plan and underlying analytic code are available from the authors upon request, understanding that the computer programs may reply upon coding templates or macros that are unique to ICES and are therefore either inaccessible or may require modification.

Reviewer #1: Yes

4. Is the manuscript presented in an intelligible fashion and written in standard English?

Reviewer #1: Yes

5. Review Comments to the Author

Reviewer #1: This manuscript is technically sound and overall easy to read. The authors should be congratulated. I have no concerns regarding the conduct of the study. Some specific recommendations regarding the write-up are included below.

Title:

-For clarity, consider revising to, "The association between gender and persistent opioid use..."

Done

Ethics statement:

-There is no explicit statement that the Queen's IRB approved the research or what type of consent was obtained.

Changed in the Methods

Data availability section:

-Please supply details regarding where/how all raw study data can be obtained (even though it seems it is publicly available).

The dataset from this study is held securely in coded form at ICES. While legal data sharing agreements between ICES and data providers (e.g., healthcare organizations and government) prohibit ICES from making the dataset publicly available, access may be granted to those who meet pre-specified criteria for confidential access, available at www.ices.on.ca/DAS (email: das@ices.on.ca). The full dataset creation plan and underlying analytic code are available from the authors upon request, understanding that the computer programs may reply upon coding templates or macros that are unique to ICES and are therefore either inaccessible or may require modification.

Abstract:

-In the first sentence of the results, please indicate some statistical measure (with error term) regarding whether the unadjusted proportion of males and females with persistent opioid use are different. (Please make the same clarification in the main Results section.)

The unadjusted OR between males and females can be found in Figure 1 3-6 months (1.11 95% CI 1.05-1.17) and has been added to the results of the Abstract and in the Results section. 

-In the conclusion, consider adding a term such as, "females are at a 'slightly' higher risk..." since the prevalences of 8.7% and 9.6% (and the OR of 1.07) are, arguably, fairly similar from a clinical standpoint.

As above, although these ORs after controlling for key confounders are low, the relevance of these observations is pointed out in the Interpretation. We have qualified the conclusions as suggested as it is true that the remaining effect size is small at least in the 3-6 month timeframe. 

Results:

-When discussing the prevalence of longer term opioid use, please provide statistical metrics for the comparisons of proportions.

The ORs for the 1-2 year time frame were found in Figure 3: unadjusted 1.35 (95% CI 1.21, 1.50) and in the fully adjusted model 1.28 (95% CI 1.14, 1.43). This has been added to the Results section as well as the Figure. 

Figure 1:

-The text in the figures is quite small. Consider splitting the figure across two pages for easier readability. Or, if color-coded, you could potentially put the male and female data on the same plot.

We are hoping the quality of the figure is sufficient as we were specifically wanting to exhibit the data one on top of the other to help the reader assess the effect size for the covariates and make “comparisons” between males and females. We tried combining on the same plot but we felt this was more complicated than the larger figure with two plots on top of each other. We do have examples of this in the literature which we felt represented the observations quite well and wanted to re-capitulate. 

Figure 2:

-This is a great figure. Very digestable and informative.

Figure 3:

-Also a great figure. Consider adding an adjustment for prescription of opioids 3-6 months after index.

We have of course stratified only to those that were prescribed opioids initially and we are not sure (given the dwindling numbers of cases) if adjusting for those that received opioids in early time points (3-6 months) will be additive to the observations we are trying to convey. 

Other:

-Both gender and sex are mentioned in the Intro, but then the manuscript seems to focus on only gender. What was the rationale behind focusing on gender as opposed to sex (since they are not interchangeable variables)? Does the administrative database truly capture gender, sex, or both?

Thank you for this comment and we are quite cognizant of the difference between the terminology and agree with the SAGER guidelines. With the recent additions of gender identity and gender expression to the Canadian Human Rights Act and the Criminal Code as well as some sources of administrative data changing from sex to gender, it is of course necessary to distinguish the concepts of sex, gender identity and gender expression. Statistics Canada has revised the variable 'sex of person' as well as creating a new variable, 'gender of person'. 

Unfortunately, the dataset utilized through ICES would be self-reported in general and different options to capture gender identity other than binary (male/female) is not possible. However it would also be true that biological sex would not be specifically captured within the administrative data. Although both sex (biological) and gender (socially constructed roles, behaviours, expressions) may play a role in in the primary outcome of this study we felt that as the data point is self-reported, and as some of the covariates/concepts more align with socially determined health care utilization/interactions, that gender would be the preferred term for this study. As mentioned we did discuss this briefly in the Introduction and were conscientious to use the term gender as well as male/female throughout. 

6. PLOS authors have the option to publish the peer review history of their article (what does this mean?). If published, this will include your full peer review and any attached files.

Do you want your identity to be public for this peer review? For information about this choice, including consent withdrawal, please see our Privacy Policy.

Reviewer #1: No

---

## [Decision Letter · Decision Letter 1]

9 Jul 2021

PONE-D-21-08670R1

The Association of Gender and Persistent Opioid Use Following an Acute Pain Event: A Retrospective Population Based Study of Renal Colic

PLOS ONE

Dear Dr. Siemens,

Thank you for submitting your manuscript to PLOS ONE. After careful consideration, we feel that it has merit but does not fully meet PLOS ONE’s publication criteria as it currently stands. Therefore, we invite you to submit a revised version of the manuscript that addresses the points raised during the review process.

We look forward to receiving your revised manuscript.

Kind regards,

Jingjing Qian

Academic Editor

PLOS ONE

Journal Requirements:

Additional Editor Comments (if provided):

Thanks for addressing majority of comments. Due to the small/modest adjusted difference between male and female found in the study, we would like the authors to take additional edits before we can accept your paper. Specifically:

1. Abstract:"Interpretation: After controlling for key covariates, females are at higher risk of demonstrating long term opioid use following an episode of renal colic. Evidence of prior mental health service utilization and acute colic care did not appear to significantly explain these observations." Please add "slightly" before "higher risk of demonstrating long term opioid use following an episode of renal colic" as you did in the conclusion section.

2. Methods: page 7, covariates -- the authors provided appropriate explanation in the responses to comments document, but did not incorporate corresponding edits in this section to justify selection of covariates. Please make the edits and cite relevant studies.

Reviewers' comments:

Reviewer's Responses to Questions

**Comments to the Author**

1. If the authors have adequately addressed your comments raised in a previous round of review and you feel that this manuscript is now acceptable for publication, you may indicate that here to bypass the “Comments to the Author” section, enter your conflict of interest statement in the “Confidential to Editor” section, and submit your "Accept" recommendation.

Reviewer #1: All comments have been addressed

2. Is the manuscript technically sound, and do the data support the conclusions?

Reviewer #1: Yes

3. Has the statistical analysis been performed appropriately and rigorously? 

Reviewer #1: Yes

4. Have the authors made all data underlying the findings in their manuscript fully available?

Reviewer #1: No

5. Is the manuscript presented in an intelligible fashion and written in standard English?

Reviewer #1: Yes

6. Review Comments to the Author

Reviewer #1: (No Response)

7. PLOS authors have the option to publish the peer review history of their article (what does this mean?). If published, this will include your full peer review and any attached files.

Reviewer #1: No

---

## [Author Response · Author response to Decision Letter 1]

5 Aug 2021

May 26, 2021

Jingjing Qian

Academic Editor

PLOS ONE

Dear Dr. Qian:

Please find enclosed our manuscript re-submission (revision 2) entitled “The Association of Gender and Persistent Opioid Use Following an Acute Pain Event: A Population Based Study of Renal Colic”. Thank you for the opportunity to re-submit a revised version of this manuscript for consideration of publication in your journal. 

Sincerely,

D. Robert Siemens

Professor and Chair

Department of Urology

Queen's University

 

Editors and Reviewers Comments: 

Thanks for addressing majority of comments. Due to the small/modest adjusted difference between male and female found in the study, we would like the authors to take additional edits before we can accept your paper. Specifically:

1. Abstract:"Interpretation: After controlling for key covariates, females are at higher risk of demonstrating long term opioid use following an episode of renal colic. Evidence of prior mental health service utilization and acute colic care did not appear to significantly explain these observations." Please add "slightly" before "higher risk of demonstrating long term opioid use following an episode of renal colic" as you did in the conclusion section.

This is very much appropriate and done (page 3)

2. Methods: page 7, covariates -- the authors provided appropriate explanation in the responses to comments document, but did not incorporate corresponding edits in this section to justify selection of covariates. Please make the edits and cite relevant studies.

 Thank you. We have added a few sentences in the Methods explaining what we discussed in the rebuttal letter (with references). 

Journal Requirements:

Further there was this one comment by a reviewer and we previously have addressed giving the ICES privacy policy. We are happy to add this as some supplementary comment or in “Acknowledgements”?

4. Have the authors made all data underlying the findings in their manuscript fully available?

No

The dataset from this study is held securely in coded form at ICES. While legal data sharing agreements between ICES and data providers (e.g., healthcare organizations and government) prohibit ICES from making the dataset publicly available, access may be granted to those who meet pre-specified criteria for confidential access, available at www.ices.on.ca/DAS (email: das@ices.on.ca). The full dataset creation plan and underlying analytic code are available from the authors upon request, understanding that the computer programs may reply upon coding templates or macros that are unique to ICES and are therefore either inaccessible or may require modification.

---

## [Editor Report · Decision Letter 2]

11 Aug 2021

The Association of Gender and Persistent Opioid Use Following an Acute Pain Event: A Retrospective Population Based Study of Renal Colic

PONE-D-21-08670R2

Dear Dr. Siemens,

We’re pleased to inform you that your manuscript has been judged scientifically suitable for publication and will be formally accepted for publication once it meets all outstanding technical requirements.

Kind regards,

Jingjing Qian

Academic Editor

PLOS ONE

Additional Editor Comments (optional):

Thanks for addressing all comments and making edits as suggested.
---

## [Editor Report · Acceptance letter]

19 Aug 2021

PONE-D-21-08670R2 

The Association of Gender and Persistent Opioid Use Following an Acute Pain Event: A Retrospective Population Based Study of Renal Colic 

Dear Dr. Siemens:

I'm pleased to inform you that your manuscript has been deemed suitable for publication in PLOS ONE. Congratulations! Your manuscript is now with our production department. 

Kind regards, 

on behalf of

Dr. Jingjing Qian 

Academic Editor

PLOS ONE